# Sensory Processing Sensitivity and Gastrointestinal Symptoms in Japanese Adults

**DOI:** 10.3390/ijerph19169893

**Published:** 2022-08-11

**Authors:** Shuhei Iimura, Satoshi Takasugi

**Affiliations:** 1Soka University, 1-236 Tangi-machi, Hachioji, Tokyo 192-8577, Japan; 2R&D Division, Meiji Co., Ltd., 1-29-1 Nanakuni, Hachioji, Tokyo 192-0919, Japan

**Keywords:** sensory processing sensitivity, environmental sensitivity, highly sensitive person, gastrointestinal symptoms, physical health

## Abstract

Background: Sensory processing sensitivity is a personality or temperamental trait defined as individual differences in the tendency to perceive and process both positive and negative stimuli and experiences. Studies have shown that high sensitivity is correlated with psychosocial health, including depression and anxiety. However, its relationship with physical health has not been clarified. To fill this gap, using a large sample size with sufficient statistical power, an adult sample not including university students, and a range of covariates, this study examined the association between gastrointestinal symptoms as an indicator of physical health and sensory processing sensitivity. Methods: In this cross-sectional study, the participants were 863 Japanese adults (female = 450; male = 413; *M*_age_ = 30.4 years; *SD* = 4.9) who completed a web-based questionnaire. We statistically controlled for sociodemographic characteristics and examined whether sensory processing sensitivity is correlated with gastrointestinal symptoms. Results: The results showed that highly sensitive individuals were more likely to experience a wide range of gastrointestinal symptoms in the past week, including reflux symptoms, abdominal pain, indigestion symptoms, diarrhea symptoms, and constipation symptoms, even when statistically controlling for the participants’ sociodemographic characteristics. Conclusions: Our findings suggest that high sensory processing sensitivity is associated with physical health. Some of the potential causes of this are also discussed.

## 1. Introduction

People differ in their sensitivity to both positive and negative internal and external stimuli [1]. Some people are more susceptible to both supportive and adversarial experiences than others. Importantly, individual differences in sensitivity are related to the socio-emotional well-being in our daily lives [2,3], but the relationship with physical health, which is another important aspect of our well-being, remains unclear. To fill this gap, this study sought to focus on gastrointestinal disease symptoms as an indicator of physical health and examined their relationship to sensitivity.

### 1.1. Individual Differences in Sensory Processing Sensitivity

Susceptibility to environmental influences is not a binary variable, but a continuous variable with an assumed normal distribution, ranging from low to high [4]. According to several prevailing theories [5,6], the interaction between susceptibility genes [7] and the early environment (both supportive and adversarial) [8] shapes the susceptibility of the central nervous system. When highly sensitive individuals are exposed to certain environmental stimuli, an increase in neurophysiological reactivity is observed (e.g., activation of the amygdala and insular cortex; [9]), along with behaviors exhibiting heightened sensitivity (e.g., difficult temperament; [10]).

Sensory processing sensitivity can be defined as a combination of personality or temperamental characteristics that explain differences in individual sensitivity to positive and negative experiences and environments [1]. This concept represents individual differences in the tendencies to perceive and process environmental stimuli and is characterized by greater depth of information processing, heightened empathic and emotional reactivity, increased awareness of environmental subtleties, and becoming overwhelmed more easily [11]. Researchers in this area have assessed individual differences in sensory processing sensitivity (SPS) using the highly sensitive person (HSP) scale, a self-reported questionnaire [12]. The label “HSP” is used for individuals with a relatively high SPS.

Researchers in this area have noted that SPS is a concept that correlates with, but is distinct from, some of the other personality traits. For example, a recent meta-analysis reported that SPS in adults was moderately and positively associated with neuroticism and was unrelated to the other Big Five personality traits, including extroversion, openness, agreeableness, and conscientiousness [13]. In addition, a behavioral genetic study using a large sample of adolescent twins estimated that 20% of the heritability of SPS is due to genetic influences specific to sensitivity that are not explained by the Big Five personality traits [14].

Several studies have pointed out that SPS correlates with mental health, as do other personality traits, such as neuroticism. For example, a cross-sectional study by Liss et al. [2,3] that used a sample of American university students reported that SPS was positively correlated with depressive symptoms, anxiety, and alexithymia. In addition, a study by Iimura [15] that used a sample of Japanese university students found that those with higher SPS were more likely to report coronavirus disease 2019 (COVID-19)-related distress. In a study that used short-term longitudinal data, highly sensitive youths reported that their mental health was more likely to be affected, for better or worse, by life events that they experienced in the previous week [16].

### 1.2. Sensory Processing Sensitivity and Gastrointestinal Symptoms

There is a need for more studies focusing on physical health, to understand the well-being of sensitive people from both psychological and physical aspects. Unfortunately, while findings suggesting a link between SPS and mental health are accumulating, the relationship between individuals with high sensitivity and physical health has not yet been fully reported, with a few exceptions [17,18,19]. In one of the few findings, Behnam [17] examined the association between SPS and physical symptoms—including back pain, diarrhea, heartburn, and sore throat—in American university students. The results showed that HSP scale scores were positively associated with somatic symptoms, even after controlling for sex. In contrast, Grimen and Diseth [18] reported that Norwegian university students did not show a clear correlation between SPS and somatic symptoms. Furthermore, Takahashi et al. [19] reported a positive association between two of the HSP subscales (ease of excitation and low sensory threshold) and physical symptoms—including cardiovascular, respiratory, and gastrointestinal—in Japanese youths.

### 1.3. The Current Study

Previous studies have focused on the mental health of highly sensitive persons and have overlooked their association with physical health. As proposed for the agenda of the research area on *Environmental Sensitivity* [1], the role of SPS in physical health is still not well understood and clarifying this would also be helpful when considering support for those who are susceptible to distress in stressful situations. Hence, the present study focuses on gastrointestinal symptoms as an indicator of physical health and explores their relationship with SPS. Among our various indicators of physical symptoms, we address gastrointestinal symptoms for two main reasons. First, although findings are limited, previous studies of SPS have often examined its relationship to physical symptoms, particularly gastrointestinal symptoms [17,18,19]. Therefore, by focusing on gastrointestinal symptoms in this study as well, an indirect comparison with previous studies can be made. Second, a common underpinning factor for SPS and gastrointestinal symptoms is the serotonin (5-HT). Although findings are limited, individual differences in sensitivity to environmental influences have been suggested to be associated with the serotonin transporter gene polymorphism (5-HTTLPR) [5]. In addition, approximately 90% of serotonin in the body is produced in the intestinal enterochromaffin cells, which are involved in gastrointestinal tract functions (e.g., electrolyte absorption, fluid homeostasis, gastrointestinal motility, and gut permeability) [20].

Although there has been some evidence to date of a link between SPS and physical health, several issues need to be addressed. The first issue is sample bias. Existing findings depend primarily on data obtained from younger samples, including students in college [17,18,19]. To address this issue, we obtained data from a sample of non-students aged 20–39. The second issue is that some findings report a positive correlation between sensory processing sensitivity and physical health, while others do not, thereby showing an inconsistency. A potential reason for this could be a lack of statistical power (e.g., Grimen and Diseth obtained data from only 169 students, including 28 male students). This study addressed this issue by obtaining data from 863 individuals (52.1% female) with sufficient statistical power. The third issue is that previous studies have used limited control variables, including sex and age [17,19]. Epidemiological studies assessing the physical health status of individuals typically statistically control for a variety of relevant factors, including education, annual income, frequency of alcohol consumption, smoking habits, physical activity, and body mass index. Therefore, the present study collected data on a wide range of factors and statistically controlled for their effects.

### 1.4. Hypothesis

Given that the traits of SPS are characterized by low sensory thresholds and the likelihood of being easily overwhelmed, this study predicted that highly sensitive individuals would report higher levels of gastrointestinal symptoms.

## 2. Materials and Methods

### 2.1. Procedure and Participants

This cross-sectional study involved a web-based survey. We recruited Japanese adults aged 20–39 years through a social survey company (Macromill, Inc., Tokyo, Japan). As shown in Figure 1, 1866 individuals expressed interest in participating in this study; however, individuals who could not give informed consent (*N* = 215), individuals residing outside of Japan (*N* = 7), students (*N* = 197), and individuals over 40 years old (*N* = 4) were excluded from this study. To address the issue that existing findings are based primarily on university student samples [17,18,19], in this study, we recruited non-student adults. After excluding 423 participants who did not meet the inclusion and exclusion criteria, 1443 participants took part in the survey. Participants answered questions about sociodemographic characteristics and psychological scales using a web-based survey form. The analysis excluded 388 participants (26.9%) who did not complete the survey and 55 participants (3.8%) who gave inappropriate responses (e.g., answered too fast). As a result of an attention check using the Directed Questions Scale [21], 137 participants (9.5%) were excluded from the analysis because they chose other options for the item “Please select not at all for this item.” Only 863 participants (female = 450 [52.1%]; male = 413 [47.9%]) were included for the final analysis. The mean age of the participants was 30.4 years (*SD* = 4.9). The sociodemographic characteristics of the participants, such as household income and education level, are shown in Table 1. The analyzed data did not include any missing values.

### 2.2. Ethical Statement

This study was approved by the Ethics Committee of Meiji Co., Ltd (Tokyo, Japan). Institutional Review Board (No. 2020-005), considering the guidelines from the Declaration of Helsinki.

### 2.3. Measures

#### 2.3.1. Sensory Processing Sensitivity

The Japanese version of the HSP scale [22] was used to measure individual differences in SPS. This scale is a Japanese translation of the 27-item English version and consists of 19 items that measure the sensitivity of Japanese adults based on the results of exploratory factor analysis. Similar to the English version, the Japanese version consists of three factors (ease of excitation, low sensory threshold, and aesthetic sensitivity). However, in this study, the scale score, which is the average of all the items, was used for the analysis. The scale score reflects individual differences in sensitivity to both positive and negative stimuli [23]. This scale includes items such as “Are you easily overwhelmed by strong sensory input?” and “Are you deeply moved by the arts or music?” Each item was rated on a 7-point Likert-type scale, ranging from 1 (strongly disagree) to 7 (strongly agree). The internal consistency was excellent, with a Cronbach’s α of 0.90.

#### 2.3.2. Gastrointestinal Symptoms

The 15-item Japanese version of the Gastrointestinal Symptom Rating Scale [24] was used to measure gastrointestinal symptoms. This scale is composed of five subscales, including reflux symptoms (e.g., Have you been bothered by heartburn during the past week?), abdominal pain (e.g., Have you been bothered by stomach ache or pain during the past week?), indigestion symptoms (e.g., Have you been bothered by rumbling in your stomach or belly during the past week?), diarrhea symptoms (e.g., Have you been bothered by diarrhea during the past week?), and constipation symptoms (e.g., Have you been bothered by constipation during the past week?), which is similar to the original scale [25,26]. Participants self-reported the gastrointestinal symptoms that they experienced during the previous week using a 7-point Likert-type scale (1 = no discomfort at all, 7 = very severe discomfort). Internal consistency was α = 0.80 for reflux symptoms, α = 0.79 for abdominal pain, α = 0.83 for indigestion symptoms, α = 0.87 for diarrhea symptoms, and α = 0.82 for constipation symptoms, respectively.

#### 2.3.3. Control Variables

As possible covariates suggested to be associated with gastrointestinal symptoms, (e.g., [27,28]), the sociodemographic characteristics, as shown in Table 1, were used as control variables in the analysis of this study. For example, gastrointestinal symptoms are associated with factors such as income [29], alcohol consumption [30], smoking habits [31], BMI [32], allergies [33], previous experience with therapy [34], and physical activity [35].

### 2.4. Data Analysis

Our analysis plan consisted of two main parts. The first was a preliminary analysis that calculated the descriptive statistics and correlation coefficients between SPS and gastrointestinal symptoms. The second was a hierarchical multiple regression analysis with SPS as the independent variable and the five subscales of gastrointestinal symptoms as the dependent variables. In this analysis, we controlled for sociodemographic characteristics (Step 1) and examined whether SPS significantly explained gastrointestinal symptoms (Step 2). If *R*^2^ increased significantly from Step 1 to Step 2 (Δ*R*^2^), then SPS could be interpreted as a factor explaining the gastrointestinal symptoms. The statistical significance test was repeated for each of the five subscales of gastrointestinal symptoms. The significance level (α) in this study was corrected by the Bonferroni method, and *p* < 0.01 (0.05/5 = 0.01) was considered statistically significant. All the data in this study were analyzed with R version 3.6.3 (R Core Team: Vienna, Austria) [36]. The R code used for this study appears in the Open Science Framework (OSF; https://bit.ly/3FstaAn, accessed on 7 August 2022).

## 3. Results

### 3.1. Preliminary Analysis

Table 1 presents the means and standard deviations of the SPS and gastrointestinal symptoms. No ceiling or floor effects were observed for any of the variables. Appendix A, which have been uploaded to the OSF, show the histograms for each variable. In addition, the SPS and gastrointestinal symptoms characteristics based on sociodemographic characteristics are available on the OSF.

SPS was weakly positively correlated with all five subscales of gastrointestinal symptoms (Table 2). Specifically, it was associated with reflux symptoms *r* = 0.19 (95% Confidence Interval {CI} [0.12, 0.25], *p* < 0.001), abdominal pain *r* = 0.24 (95% CI [0.17, 0.30], *p* < 0.001), indigestion symptoms *r* = 0.25 (95% CI [0.19, 0.31], *p* < 0.001), diarrhea symptoms *r* = 0.20 (95% CI [0.14, 0.27], *p* < 0.001), and constipation symptoms (*r* = 0.26 (95% CI [0.20, 0.32], *p* < 0.001). Correlation coefficients between all variables are uploaded to OSF as Appendix A. Regarding these coefficients, statistical power analysis, conducted using G*Power [37], confirmed that the current sample size had sufficient statistical power (power [1 − β] = 0.99) to detect *r* > |0.19|.

### 3.2. Regression Analysis

The results for each of the five subscales of gastrointestinal symptoms are shown in Table 3. Even after controlling for participants’ sociodemographic characteristics, the results suggested that SPS was a significant predictor of all domains of gastrointestinal symptoms. Specifically, those with higher SPS were more likely to have reflux symptoms (*b* = 0.25, β = 0.21, 95% CI [0.15, 0.28], *p* < 0.001), abdominal pain (*b* = 0.29, β = 0.24, 95% CI [0.18, 0.31], *p* < 0.001), indigestion symptoms (*b* = 0.27, β = 0.25, 95% CI [0.18, 0.32], *p* < 0.001), diarrhea symptoms (*b* = 0.29, β = 0.23, 95% CI [0.16, 0.30], *p* < 0.001), and constipation symptoms (*b* = 0.30, β = 0.24, 95% CI [0.18, 0.31], *p* < 0.001). In addition, the *R*^2^ significantly increased from Step 1 to Step 2 in all areas of gastrointestinal symptoms (Δ*R*^2^ = 0.04–0.11). A plot of the regression line showing the relationship between SPS, and the five subscales of gastrointestinal symptoms can be found in the Appendix A, Appendix B, Appendix C, Appendix D and Appendix E.

## 4. Discussion

Researchers in SPS have tried to accumulate knowledge about the mental health of sensitive persons [2,3]. However, to further understand the well-being of susceptible individuals, it is necessary to not only identify the psychosocial aspects, but also their physical health characteristics. Unfortunately, two decades of research on sensitivity have overlooked the consideration of physical health. Therefore, to fill a gap in the existing research, this study focused on five gastrointestinal symptoms as indicators of physical health and exploratively examined their relationship to SPS. The results showed that greater sensitivity was weakly positively associated with higher levels of a wide range of gastrointestinal symptoms. These symptoms included reflux symptoms, abdominal pain, indigestion symptoms, diarrhea symptoms, and constipation symptoms. This result occurred even after statistically controlling for participants’ sociodemographic characteristics.

### 4.1. Why Did Highly Sensitive Persons Report Gastrointestinal Symptoms?

Why did SPS positively correlate with self-reported gastrointestinal symptoms? To answer this question, this study proposes several interpretations.

First, SPS is characterized by low sensory thresholds and easier excitation [1], which may make it easier to notice slight pain or discomfort inside the body. Our results may be explained by the interoception, the sensitivity to visceral sensations. However, being able to easily notice one’s own gastrointestinal symptoms is not necessarily associated with the development of gastrointestinal diseases. We wish to emphasize that this study cannot make any claims about the actual development of gastrointestinal diseases, because this study merely examined the association between SPS and self-reported gastrointestinal symptoms.

Second, highly sensitive individuals are more likely to be adversely affected by daily hassles and recent negative life events [16]. They may also be more prone to developing gastrointestinal symptoms due to stressful environmental factors. Certain stressful life events have been associated with the onset or worsening of symptoms of common gastrointestinal disorders, including functional gastrointestinal disorders, inflammatory bowel disease, gastroesophageal reflux disease, and peptic ulcer disease [38]. Therefore, it is possible that heightened sensitivity may strengthen the relationship between stressful life events and gastrointestinal symptoms.

Third, given the recent advances in understanding gut–brain interactions [39,40], highly sensitive individuals may have a neural basis associated with the perception or development of gastrointestinal symptoms. For example, a review by Mayer and Tillisch [40] reported that in a study examining the brain responses to controlled rectal balloon distension, patients with IBS showed more activity in the brain regions associated with stress and arousal circuits than in healthy controls. In addition, a recent study examining emotional stimuli and brain reactivity [9] suggested that highly sensitive individuals are more likely to have activation of the amygdala, which is associated with reactivity to stress. Focusing on the activation of brain regions related to stress and arousal circuits, there may be a neurophysiological link between high sensitivity and brain–gut interaction.

Fourth, SPS is suggested to be associated with serotonin transporter (5-HTT)-linked polymorphic region (5-HTTLPR) [41]. Serotonin (5-HT) is a monoamine neurotransmitter in both the central nervous system and gastrointestinal tract. Approximately 90% of the 5-HT in the body is produced in the enterochromaffin cells, and plays various roles in the gastrointestinal tract (e.g., electrolyte absorption, fluid homeostasis, gastrointestinal motility, and gut permeability) [20]. The intestinal 5-HTT plays an important role in the clearance of 5-HT [42,43], while 5-HTTLPR affects the activity of 5-HTT [44]. Given these facts, gastrointestinal symptoms in highly sensitive individuals may be closely associated with the activity of intestinal 5-HTT. Unfortunately, there is little empirical evidence of this relationship (e.g., Licht et al. [45] denied a clear association between the two), and further studies are required.

### 4.2. Strengths, Limitations, and Future Directions

The strength of the current study is that it is the first to provide evidence of an association between SPS and gastrointestinal symptoms in a large sample of Japanese adults. In other words, our findings suggest that individual differences in SPS are not only associated with psychosocial health but also with physical health. However, many issues still need to be resolved to gain a deeper understanding of the relationship between sensitivity and physical health. The limitations of this study are as follows.

First, because the data obtained in this study are based on self-reports, it cannot be concluded that highly sensitive individuals have diseases of gastrointestinal function. To investigate this, it may be necessary to compare the sensitivity scores of IBS patients and healthy controls, or to compare highly sensitive and less sensitive persons for gut microbiota characteristics associated with gastrointestinal symptoms. Second, this study statistically controlled for a wide range of participants’ sociodemographic characteristics but did not control for the effects of environmental variables such as psychosocial stressors that may be associated with gastrointestinal symptoms. This may be related to the small coefficient of determination (*R*^2^) for gastrointestinal symptoms in our analysis. Future studies can expand the knowledge base by examining individual differences in gastrointestinal symptoms in terms of sensitivity and environmental interactions. Third, future research can further clarify the role or uniqueness of SPS in gastrointestinal symptoms by statistically controlling for interoception and anxiety sensitivity, which are related to visceral sensation perception. Fourth, the findings of this study are based on data obtained from a web survey and may have been affected by sampling bias, including selection bias and reporter bias. Finally, unfortunately, there is currently insufficient elaboration of theories linking SPS and physical health. For example, is SPS involved in the perception of gastrointestinal symptoms via interoception, the sensitivity to visceral sensations? How does it relate to actual gastrointestinal disorders? What factors are involved in its neural and physiological basis and mechanisms? These questions will be the agenda to be resolved to further understand the role of SPS in physical health.

## 5. Conclusions

Researchers have suggested that SPS, defined as the individual differences in the tendency to perceive and process positive and negative stimuli and experiences, is associated with psychosocial health. However, their relationship with physical health has been overlooked. Few previous studies have reported consistent findings on the association between SPS and gastrointestinal symptoms. Moreover, they had small sample sizes, primarily used university student samples, or did not adequately control for covariates. Therefore, to address these issues, the present study examined the correlation between SPS and gastrointestinal symptoms using a large sample size with sufficient statistical power, a sample of adults excluding university students, and a wide range of covariates. As a result, even after statistically controlling for participants’ sociodemographic characteristics, those with higher SPS were more likely to self-report gastrointestinal symptoms, including reflux symptoms, abdominal pain, indigestion symptoms, diarrhea symptoms, and constipation symptoms in the past week. Our findings suggest that SPS is associated with physical health.

## Figures and Tables

**Figure 1 ijerph-19-09893-f001:**
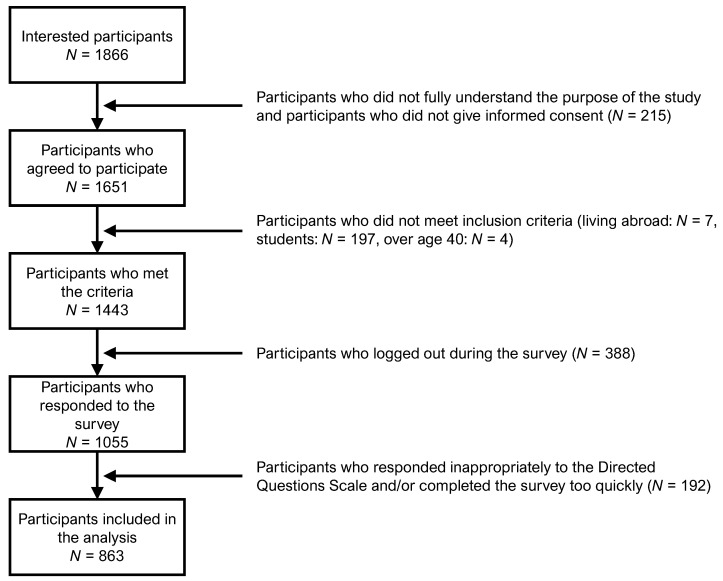
Procedures for recruiting participants in this study.

**Table 1 ijerph-19-09893-t001:** Sample Characteristics (*N* = 863) ^1^.

	*N* (%)
** *Gender* **	
Female	450 (52.1%)
Male	413 (47.9%)
** *Major and professional experience in nutrition and health* **	
Yes	114 (13.2%)
No	749 (86.8%)
** *Number of family members living together* **	
One	191 (22.1%)
Two	167 (19.4%)
Three	234 (27.1%)
Four	180 (20.9%)
Five	61 (7.1%)
Six	16 (1.9%)
More than seven	14 (1.6%)
** *Marriage status* **	
Married	347 (40.2%)
Other	516 (59.8%)
** *Number of children living together* **	
None	590 (68.4%)
One	142 (16.5%)
Two	94 (10.9%)
Three	31 (3.6%)
More than four	6 (0.7%)
** *Annual household income* **	
Less than 2,000,000 JPY (≠17,505 USD)	84 (9.7%)
2,000,000 (≠17,505 USD)~4,000,000 JPY (≠35,010 USD)	231 (26.8%)
4,000,000 (≠35,010 USD)~6,000,000 JPY (≠52,517 USD)	243 (28.2%)
6,000,000 (≠52,517 USD)~8,000,000 JPY (≠70,017 USD)	142 (16.5%)
8,000,000 (≠70,017 USD)~10,000,000 JPY (≠87,521 USD)	75 (8.7%)
10,000,000 (≠87,521 USD)~12,000,000 JPY (≠105,026 USD)	37 (4.3%)
More than 12,000,000 JPY (≠105,026 USD)	51 (5.9%)
** *Frequency of alcohol consumption* **	
None	440 (51.0%)
1~3 days/month	137 (15.9%)
1~2 days/week	141 (16.3%)
3~4 days/week	59 (6.8%)
5~7 days/week	86 (10.0%)
** *Smoking habit* **	
None	634 (73.5%)
Used to smoke	70 (8.1%)
Currently smoking	159 (18.4%)
** *Dietary advice and therapy* **	
No dietary advice and/or therapy received	830 (96.2%)
Used to receive dietary advice and/or therapy	17 (2.0%)
Currently receiving dietary advice and/or therapy	16 (1.9%)
** *Physical activity level* **	
Low	245 (28.4%)
Moderate	503 (58.3%)
High	115 (13.3%)
** *Education* **	
Elementary and middle schools	25 (2.9%)
High school	249 (28.9%)
Junior college	158 (18.3%)
University	399 (46.2%)
Graduate school	32 (3.7%)
** *Food allergy* **	
Yes	812 (94.1%)
No	51 (5.9%)

^1^ Although not shown in Table 1, this study also calculated the participants’ body mass index (BMI). The mean BMI was 21.08 (*SD* = 3.55) for female and 22.49 (*SD* = 3.93) for male participants.

**Table 2 ijerph-19-09893-t002:** Descriptive Statistics and Correlation Coefficients among Variables (*N* = 863).

	*M*	*SD*	Kurtosis	Skewness	1	2	3	4	5
1. Sensory processing sensitivity	4.18	0.91	0.62	−0.12	-				
2. Reflux symptoms	1.70	1.05	4.62	1.99	0.19 ***	-			
3. Abdominal pain	1.85	1.08	2.94	1.62	0.24 ***	0.76 ***	-		
4. Indigestion symptoms	1.90	0.99	4.41	1.80	0.25 ***	0.63 ***	0.71 ***	-	
5. Diarrhea symptoms	1.98	1.17	2.51	1.50	0.20 ***	0.56 ***	0.62 ***	0.65 ***	-
6. Constipation symptoms	2.07	1.11	1.51	1.21	0.26 ***	0.47 ***	0.53 ***	0.60 ***	0.62 ***

*** *p* < 0.001.

**Table 3 ijerph-19-09893-t003:** (a). Regression Models Predicting Gastrointestinal Symptoms (Reflux Symptoms and Abdominal Pain). (b). Regression Models Predicting Gastrointestinal Symptoms (Indigestion Symptoms and Diarrhea Symptoms). (c). Regression Models Predicting Gastrointestinal Symptoms (Constipation Symptoms).

**(a)**												
	**Reflux Symptoms**						**Abdominal Pain**					
	**Step 1**			**Step 2**			**Step 1**			**Step 2**		
	** *b* **	** *SE* **	**β**	** *B* **	** *SE* **	**β**	** *b* **	** *SE* **	**β**	** *b* **	** *SE* **	**β**
Gender	0.18	0.08	0.08	0.12	0.08	0.06	0.40 ***	0.08	0.19	0.34 ***	0.08	0.16
Age	−0.01	0.01	−0.05	−0.01	0.01	−0.06	−0.02	0.01	−0.07	−0.02	0.01	−0.09
Specialty ^1^	−0.20	0.11	−0.06	−0.20	0.11	−0.06	−0.18	0.11	−0.06	−0.18	0.11	−0.06
Family ^2^	−0.01	0.03	−0.02	−0.02	0.03	−0.02	0.03	0.03	0.04	0.03	0.03	0.04
Marriage ^3^	−0.11	0.09	−0.05	−0.19	0.09	−0.09	−0.16	0.09	−0.08	−0.26 **	0.09	−0.12
Children ^4^	0.00	0.06	0.00	0.02	0.06	0.01	−0.10	0.06	−0.08	−0.08	0.06	−0.06
Income ^5^	−0.04	0.03	−0.05	−0.03	0.03	−0.05	−0.06	0.03	−0.09	−0.05	0.03	−0.08
Alcohol consumption ^6^	−0.03	0.02	−0.06	−0.04	0.02	−0.06	−0.04	0.02	−0.06	−0.04	0.02	−0.06
Smoking ^7^	−0.06	0.03	−0.07	−0.09	0.03	−0.09	−0.07	0.04	−0.08	−0.10 **	0.03	−0.11
Advice/therapy ^8^	−0.38 **	0.12	−0.11	−0.34 **	0.12	−0.10	−0.40 **	0.12	−0.11	−0.36 **	0.12	−0.10
Activity ^9^	−0.02	0.06	−0.01	−0.05	0.06	−0.03	−0.04	0.06	−0.03	−0.07	0.06	−0.04
Education	0.00	0.04	0.00	0.01	0.04	0.01	0.00	0.04	0.00	0.01	0.04	0.01
Allergy ^10^	−0.32	0.15	−0.07	−0.25	0.15	−0.06	−0.37	0.15	−0.08	−0.29	0.15	−0.06
BMI ^11^	0.02	0.01	0.07	0.02	0.01	0.08	0.01	0.01	0.04	0.01	0.01	0.05
Sensitivity ^12^				0.25 ***	0.04	0.21				0.29 ***	0.04	0.24
*R* ^2^	0.05 ***			0.09 ***			0.07 ***			0.13 ***		
*F*	3.02			5.47			4.89			8.22		
*df*	14, 848			15, 847			14, 848			15, 847		
Δ*R*^2^				0.04 ***						0.05 ***		
Δ*F*				37.97						50.88		
*df*				1, 847						1, 847		
**(b)**												
	**Indigestion Symptoms**						**Diarrhea Symptoms**					
	**Step 1**			**Step 2**			**Step 1**			**Step 2**		
	** *b* **	** *SE* **	**β**	** *B* **	** *SE* **	**β**	** *b* **	** *SE* **	**β**	** *b* **	** *SE* **	**β**
Gender	0.34 **	0.07	0.17	0.28 **	0.07	0.14	0.19	0.09	0.08	0.13	0.09	0.06
Age	−0.01	0.01	−0.06	−0.01	0.01	−0.07	−0.02	0.01	−0.06	−0.02	0.01	−0.08
Specialty ^1^	−0.11	0.10	−0.04	−0.11	0.10	−0.04	−0.05	0.12	−0.02	−0.05	0.12	−0.02
Family ^2^	−0.01	0.03	−0.02	−0.01	0.03	−0.02	0.02	0.04	0.02	0.01	0.04	0.01
Marriage ^3^	−0.01	0.09	0.00	−0.10	0.09	−0.05	0.00	0.10	0.00	−0.10	0.10	−0.04
Children ^4^	−0.07	0.06	−0.06	−0.05	0.05	−0.05	−0.02	0.07	−0.01	0.00	0.06	0.00
Income ^5^	−0.04	0.02	−0.07	−0.04	0.02	−0.06	−0.05	0.03	−0.06	−0.04	0.03	−0.05
Alcohol consumption ^6^	−0.04	0.02	−0.07	−0.04	0.02	−0.08	−0.04	0.02	−0.06	−0.04	0.02	−0.06
Smoking ^7^	−0.06	0.03	−0.06	−0.08 *	0.03	−0.09	−0.09	0.04	−0.08	−0.12 **	0.04	−0.11
Advice/therapy ^8^	−0.23	0.11	−0.07	−0.19	0.11	−0.06	−0.15	0.13	−0.04	−0.11	0.13	−0.03
Activity ^9^	0.01	0.05	0.01	−0.02	0.05	−0.01	−0.07	0.07	−0.04	−0.10	0.06	−0.06
Education	0.04	0.04	0.04	0.06	0.04	0.06	−0.06	0.04	−0.05	−0.05	0.04	−0.04
Allergy ^10^	−0.26	0.14	−0.06	−0.18	0.14	−0.04	−0.26	0.17	−0.05	−0.17	0.16	−0.03
BMI ^11^	0.02	0.01	0.06	0.02	0.01	0.07	0.03 **	0.01	0.09	0.03 **	0.01	0.10
Sensitivity ^12^				0.27 **	0.04	0.25				0.29 ***	0.04	0.23
*R* ^2^				0.11 **			0.04 **			0.09 ***		
*F*	3.55			7.07			2.49			5.34		
*df*	14, 848			15, 847			14, 848			15, 847		
Δ*R*^2^				0.06 **						0.05 ***		
Δ*F*				53.24						43.44		
*df*				1, 847						1, 847		
**(c)**												
	**Constipation Symptoms**											
	**Step 1**						**Step 2**					
	** *B* **		** *SE* **		**β**		** *b* **		** *SE* **		**β**	
Gender	0.61 **		0.08		0.27		0.55 **		0.08		0.24	
Age	−0.01		0.01		−0.04		−0.01		0.01		−0.05	
Specialty ^1^	−0.24		0.11		−0.07		−0.24		0.11		−0.07	
Family ^2^	0.05		0.03		0.06		0.04		0.03		0.05	
Marriage ^3^	−0.06		0.10		−0.03		−0.15		0.09		−0.07	
Children ^4^	−0.07		0.06		−0.06		−0.05		0.06		−0.04	
Income ^5^	−0.08 *		0.03		−0.11		−0.07 *		0.03		−0.10	
Alcohol consumption ^6^	−0.04		0.02		−0.05		−0.04		0.02		−0.06	
Smoking ^7^	−0.04		0.04		−0.04		−0.07		0.03		−0.07	
Advice/therapy ^8^	−0.21		0.12		−0.06		−0.17		0.12		−0.05	
Activity ^9^	0.08		0.06		0.04		0.05		0.06		0.03	
Education	−0.02		0.04		−0.02		−0.01		0.04		−0.01	
Allergy ^10^	−0.01		0.15		0.00		0.07		0.15		0.02	
BMI ^11^	0.00		0.01		−0.01		0.00		0.01		0.00	
Sensitivity ^12^							0.30 **		0.04		0.24	
*R* ^2^	0.10 **						0.16 **					
*F*	7.02						10.45					
*df*	14, 848						15, 847					
Δ*R*^2^							0.05					
Δ*F*							52.59					
*df*							1, 847					

^1^ Specialty = Major and professional experience in nutrition and health. ^2^ Family = Number of family members living together. ^3^ Marriage = marital status. ^4^ Children = Number of children living together. ^5^ Income = annual household income. ^6^ Alcohol consumption = Frequency of alcohol consumption. ^7^ Smoking = smoking habit. ^8^ Advice/therapy = Dietary advice and therapy. ^9^ Activity = Physical activity level. ^10^ Allergy = Food allergy. ^11^ BMI = Body Mass Index. ^12^ Sensitivity = sensory processing sensitivity. *N* = 863. * *p* < 0.05, *** p* < 0.01, *** *p* < 0.001.

## Data Availability

The data that support the findings of this study are available on reasonable request from the corresponding author.

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
