# Peer review of "Sensory Processing Sensitivity and Gastrointestinal Symptoms in Japanese Adults"

_ijerph, 2022, doi:10.3390/ijerph19169893_

Round 1
Reviewer 1 Report
The requested modifications have been satisfactorily completed.
Author Response
Thank you very much for taking the time to review our paper.
Reviewer 2 Report
Please see attached file.

Author Response
**** Reviewer#2 ****
|
Comment#1: Introduction-Are authors able to further justify why they chose gastrointestinal symptoms specifically as opposed to somatic symptoms more generally? Was there a theoretical basis for this? |
Author’s response#1:
Thank you for your question. In response to your comment, we have added why we focused on gastrointestinal symptoms.
line 107
Among our various indicators of physical symptoms, we address gastrointestinal symptoms for two main reasons. First, although findings are limited, previous studies of SPS have often examined its relationship to physical symptoms, particularly gastrointestinal symptoms [17, 18, 19]. Therefore, by focusing on gastrointestinal symptoms in this study as well, an indirect comparison with previous studies can be made. Second, a common underpinning factor for SPS and gastrointestinal symptoms is the serotonin (5-HT). Although findings are limited, individual differences in sensitivity to environmental influences has been suggested to be associated with the serotonin transporter gene polymorphism (5-HTTLPR) [5]. In addition, approximately 90% of serotonin in the body is produced in the intestinal enterochromaffin cells, which are involved in gastrointestinal tract functions (e.g., electrolyte absorption, fluid homeostasis, gastrointestinal motility, and gut permeability) [20].
|
Comment#2: Methods-Exclusion criteria are not specifically defined in the methods section. Authors discuss what the inclusion criteria is (e.g., age, non-students, etc.) but it would be helpful if the authors listed exactly what the exclusion criteria was. |
Author’s response#2:
Per your suggestion, we have added to the Methods section the characteristics of the participants we excluded from this study.
|
Comment#3: Results-Did the authors first assess correlations between socio-demographic information before adding it into the regression? It seems based on the lack of reporting of correlations between these variables that the authors chose to put all socio-demographic variables into the regression without first confirming that there were positive associations to control for. Can the authors explain why they did this? It may be helpful for readers to also see the correlations between these socio-demographic variables and GI symptoms. |
Author’s response#3:
In line with your comment, we have reported the correlation coefficients between all variables in Supplementary Materials (Table S13). In general, in psychometrics and medical statistics, possible covariates based on prior research and theory are added to the regression model, regardless of whether they are correlated with the dependent variable. Therefore, in our analysis, we also used all variables that prior research suggested were associated with the dependent variable as controlled variables in our analysis.
Table S13
Zero-Order Correlations between Variables
|
1 |
2 |
3 |
4 |
5 |
6 |
7 |
8 |
9 |
10 |
11 |
12 |
13 |
14 |
15 |
16 |
17 |
18 |
19 |
|||||||||||||||||||
|
1. Gender |
― |
||||||||||||||||||||||||||||||||||||
|
2. Age |
.00 |
― |
|||||||||||||||||||||||||||||||||||
|
3. Specialty1 |
-.16** |
.06 |
― |
||||||||||||||||||||||||||||||||||
|
4. Family2 |
.01 |
.02 |
.03 |
― |
|||||||||||||||||||||||||||||||||
|
5. Marriage3 |
-.20** |
-.27** |
.08* |
-.24** |
― |
||||||||||||||||||||||||||||||||
|
6. Children4 |
.18** |
.25** |
-.02 |
.50** |
-.59** |
― |
|||||||||||||||||||||||||||||||
|
7. Income5 |
-.03 |
-.03 |
-.06 |
.30** |
-.15** |
.02 |
― |
||||||||||||||||||||||||||||||
|
8. Alcohol consumption6 |
.20** |
-.10** |
-.02 |
.07 |
.00 |
-.01 |
-.08* |
― |
|||||||||||||||||||||||||||||
|
9. Smoking7 |
.22** |
-.10** |
-.05 |
-.05 |
.03 |
-.08* |
.02 |
.30** |
― |
||||||||||||||||||||||||||||
|
10. Advice/therapy8 |
.04 |
-.01 |
.05 |
-.01 |
-.05 |
.03 |
-.04 |
.00 |
.09* |
― |
|||||||||||||||||||||||||||
|
11. Activity9 |
.22** |
-.01 |
.02 |
-.02 |
-.05 |
.01 |
-.02 |
.08* |
.16** |
-.05 |
― |
||||||||||||||||||||||||||
|
12. Education |
-.11** |
.03 |
-.10** |
-.18** |
.01 |
-.09** |
.21** |
-.09* |
.11** |
-.01 |
-.05 |
― |
|||||||||||||||||||||||||
|
13. Allergy10 |
-.05 |
.00 |
.00 |
.00 |
.04 |
-.01 |
.06 |
-.03 |
.06 |
.03 |
.02 |
.04 |
― |
||||||||||||||||||||||||
|
14. BMI11 |
-.19** |
.12** |
.01 |
.03 |
-.01 |
-.02 |
.00 |
-.01 |
-.04 |
-.03 |
-.01 |
.02 |
-.02 |
― |
|||||||||||||||||||||||
|
15. Sensitivity12 |
.14** |
-.02 |
.00 |
-.05 |
.17** |
-.12** |
-.08* |
.10** |
.16** |
-.04 |
.11** |
-.07* |
-.07* |
-.06 |
― |
||||||||||||||||||||||
|
16. Reflux symptoms |
.07 |
-.01 |
-.08* |
-.02 |
-.05 |
.03 |
-.05 |
-.05 |
-.08* |
-.11** |
-.01 |
-.02 |
-.09** |
.06 |
.19** |
― |
|||||||||||||||||||||
|
17. Abdominal symptoms |
.16** |
-.05 |
-.09** |
.00 |
-.05 |
.01 |
-.06 |
-.02 |
-.06 |
-.11** |
.00 |
-.04 |
-.11** |
.01 |
.24** |
.76** |
― |
||||||||||||||||||||
|
18. Indigestion symptoms |
.13** |
-.05 |
-.07 |
-.07* |
.02 |
-.05 |
-.06 |
-.04 |
-.04 |
-.07* |
.03 |
.02 |
-.08* |
.03 |
.25** |
.63** |
.71** |
― |
|||||||||||||||||||
|
19. Diarrhea symptoms |
.04 |
-.04 |
-.02 |
.01 |
.01 |
.00 |
-.07 |
-.05 |
-.10** |
-.04 |
-.04 |
-.08* |
-.07* |
.07* |
.20** |
.56** |
.62** |
.65** |
― |
||||||||||||||||||
|
20. Constipation symptoms |
.28** |
-.04 |
-.11** |
.01 |
-.04 |
.03 |
-.09** |
.01 |
.01 |
-.05 |
.10** |
-.08* |
-.03 |
-.06 |
.26** |
.47** |
.53** |
.60** |
.62** |
||||||||||||||||||
1Specialty = Major and professional experience in nutrition and health. 2Family = Number of family members living together. 3Marriage = marital status. 4Children = Number of children living together. 5Income = annual household income. 6Alcohol consumption = Frequency of alcohol consumption. 7Smoking = smoking habit. 8Advice/therapy = Dietary advice and therapy. 9Activity = Physical activity level. 10Allergy = Food allergy. 11BMI = Body Mass Index. 12Sensitivity = sensory processing sensitivity. N = 863.
* p <.05, ** p <.01, *** p <.001
|
Comment#4: Discussion-In section 4.1 lines 265-266 the authors state that highly sensitive persons report more GI symptoms than less sensitive. This is an incorrect statement to make. Based on their methods/results, it is accurate to say that there is a positive relationship between SPS and GI symptoms, but that doesn’t mean that they report significantly more symptoms than those who are less sensitive. Authors should revise the statement to be more representative of the correlational nature of their analyses. I do think it’s an interesting question, though, and if authors want to make this claim, they could group the participants into two groups “highly sensitive” and “less sensitive” based on cutoff scores (if they exist) and then do a t-test between the two. |
Author’s response#4:
We have revised the wording as you indicated.
|
Comment#5: Discussion-In section 4.1 lines 268-269 the authors give one explanation of why higher SPS is correlated with more GI symptom reporting. It is an interesting and intriguing hypothesis. However, they only say one sentence without elaborating more. The authors should expand more on this point and identify potential mechanisms (e.g., visceral hypersensitivity). |
Author’s response#5:
From the perspective of the lack of elaboration of the theory linking SPS and GI symptoms, the link between SPS and visceral hypersensitivity was discussed.
line 358
Finally, unfortunately, there is currently insufficient elaboration of theories linking SPS and physical health. For example, is SPS involved in the perception of gastrointestinal symptoms via interoception, the sensitivity to visceral sensations? How does it relate to actual gastrointestinal disorders? What factors are involved in its neural and physiological basis and mechanisms? These questions will be the agenda to be resolved to further understand the role of SPS in physical health.
|
Comment#6: Discussion-In section 4.1 lines 271-273 I applaud the authors for stating that their findings do not strongly suggest a relationship between higher sensitivity and development of GI disease, as GI symptoms do not necessarily mean someone has a GI disease or disorder. However, the way it’s written makes it sound as if there was the potential to find this outcome, but the results themselves did not support the finding. Instead, the results of this study cannot make ANY claims about higher sensitivity leading to increased GI disease because 1) it’s cross-sectional and 2) it didn’t assess for GI disease, only for self-reported GI symptoms. I suggest the authors edit that sentence to be more transparent that the study cannot make any claims about the actual development of GI diseases, only self-reported symptoms. |
Author’s response#6:
We have revised the sentence as you suggested.
line 300
We wish to emphasize that this study cannot make any claims about the actual development of gastrointestinal diseases, because this study merely examined the association between SPS and self-reported gastrointestinal symptoms.
|
Comment#7: Discussion-Although there was a significant findings, correlation coefficients and R squared in the regression were quite small. Can the authors hypothesize why they found such small effects despite being well-powered and what other factors may be playing a role? |
Author’s response#7:
The background of the small coefficient of determination for gastrointestinal symptoms was mentioned in the Limitations section of this study.
|
Comment#8: Overall comments-The language around GI symptoms is a bit inconsistent. First, the authors use the term “gastrointestinal disease symptoms” but then discuss syndromes (e.g., reflux syndrome, indigestion syndrome, etc.). Gastrointestinal diseases are illnesses of organic nature, such as inflammatory bowel disease, gastroesophageal reflux disease, etc., whereas syndromes (like IBS) are considered gastrointestinal disorders. I suggest the authors remove the term “gastrointestinal disease symptoms” and just use the term “gastrointestinal symptoms.” Further, when describing the subscales of the GSRS from “reflux syndrome”, “indigestion syndrome” etc. and also just say “reflux symptoms”, “indigestion symptoms”, etc. |
Author’s response#8:
Thank you for your advice. We have changed the term to “gastrointestinal symptoms” instead of “gastrointestinal disease symptoms” as you suggested. We have also made the same correction to the GSRS subscales.
|
Comment#9: Overall comments-The authors state several times in the introduction and discussion that it is important to understand how SPS relates to physical health, but don’t expand further on why exactly it is important. Perhaps they could include a more detailed explanation of why this is an important question to tackle from both a theoretical perspective and also what the implications of this research are. |
Author’s response#9:
Thank you for your suggestion. We have added this point to the Introduction section.
line 103
As proposed for the agenda of the research area on Environmental Sensitivity [1], the role of SPS in physical health is still not well understood and clarifying this would also be helpful to consider support for those who are susceptible to distress in stressful situations.
|
Comment#10: Overall comments-There are some inconsistencies when discussing GI symptoms, diseases, disorders, etc. In addition to what I have commented on above, Section 4.1 lines 277-278 authors state that stress can worsen GI symptoms in functional GI disorders and IBS. IBS is considered a functional GI disorder. I recommend the authors ask a colleague who works in the GI field to review the manuscript and correct for these inconsistencies. |
Author’s response#10:
As you suggested, we reviewed the manuscript again to ensure that the wording was as appropriate as possible.
|
Comment#11: Overall comments-The authors discuss neuroticism in the introduction quite extensively but then did not measure it in the study. Why not? |
Author’s response#11:
In response to your comments, we have removed these statements.
Thank you very much for taking the time to review our paper.

This manuscript is a resubmission of an earlier submission. The following is a list of the peer review reports and author responses from that submission.
Round 1
Reviewer 1 Report
The aim of this article is to determine the relationship between sensory processing sensitivity and gastrointestinal disease symptoms in a sample of 863 adults.
The interest of the subject is well justified, since much has been said about the psychological aspects of these people, but little about the physical repercussions. The selection of gastrointestinal symptoms is appropriate because of their frequency in psychosomatic disorders.
A bibliography of interest for the topic is provided. The self-citations are appropriate, since they are a continuation of the same topic.
The methodology is correct, with a sufficient sample, and with the appropriate measuring instruments.
The results and discussion are correct. Limitations are reported
Suggestions for change are minor, and are as follows:
-Introduction:
The terms SPS and “highly sensitive person” are used interchangeably. The use of these terms and their differences could be better explained in the introduction.
-Although in section 1.3 “The current study” the purpose of the study and the hypothesis are explained (line 117), it would be convenient to specify more clearly and differentiate between the general objective and the specific objectives.
Methods
Specify the type of study design, both in text and abstract
Explain better the characteristics of the sample: what type of work, what level of stress does this work usually induce?
The ethical issue is not part of the study design. Move the ethical issue statement somewhere in its appropriate location. It would be convenient to add a new specific heading in MAterials and Methods about Ethical Considerations and remove the following text from 2.1. Procedure and participants (line 135)
“This study was approved by the Ethics Committee of Meiji Co., Ltd. Institutional Review Board (No. 2020-005), considering the guidelines from Declaration of Helsinki.”
Conclusions:
They are something vague and ambiguous. If the objectives are rewritten, the conclusions should be rewritten based on them.
Author Response
**** Reviewer#1 ****
We sincerely appreciate your constructive comments. In response to your comments, we have revised the manuscript for greater clarity. Our responses are provided in the comments below.
|
Comment#1: Introduction: The terms SPS and “highly sensitive person” are used interchangeably. The use of these terms and their differences could be better explained in the introduction. |
Author’s response#1:
Thank you for your comment, we have added that HSP in the Introduction is a label for people with relatively high sensory processing sensitivity (SPS).
p.2 line 54-55
The label “HSP” is used for individuals with relatively high SPS.
|
Comment#2: Introduction: Although in section 1.3 “The current study” the purpose of the study and the hypothesis are explained (line 117), it would be convenient to specify more clearly and differentiate between the general objective and the specific objectives. |
Author’s response#2:
To make the difference between the general and specific objectives clearer to the reader, (1) line breaks were added where appropriate, and (2) added subheadings for the hypotheses.
p.3 line 102-106
… Hence, the present study focuses on gastrointestinal disease symptoms as an indicator of physical health and explores their relationship with SPS.
Although there has been some evidence to date of a connection between SPS and physical health, several issues need to be addressed.
p.3 line 119
1.4. Hypothesis
Given that the traits of SPS are…
|
Comment#3: Methods: Specify the type of study design, both in text and abstract. |
Author’s response#3:
In accordance with your comments, we have added that this study used a web-based cross-sectional survey design in the Abstract and Methods section.
p.1 line 16
A cross-sectional survey design was used for this study.
p.3 line 125
This study used a cross-sectional research design with a web-based survey.
|
Comment#4: Methods: Explain better the characteristics of the sample: what type of work, what level of stress does this work usually induce? |
Author’s response#4:
Thank you for your suggestion. Unfortunately, this study did not investigate the type of work of the participants. However, we believe that this sample would not be biased toward any particular occupation, as we did not set any exclusion criteria for the type of work when recruiting participants.
|
Comment#5: The ethical issue is not part of the study design. Move the ethical issue statement somewhere in its appropriate location. It would be convenient to add a new specific heading in MAterials and Methods about Ethical Considerations and remove the following text from 2.1. Procedure and participants (line 135) “This study was approved by the Ethics Committee of Meiji Co., Ltd. Institutional Review Board (No. 2020-005), considering the guidelines from Declaration of Helsinki.” |
Author’s response#5:
We have made the following change, in response to your suggestion:
p.4 line 146
2.2. Ethical statement
This study was approved by the Ethics Committee of Meiji Co., Ltd. Institutional Review Board (No. 2020-005), considering the guidelines from the Declaration of Helsinki.
|
Comment#6: Conclusions: They are something vague and ambiguous. If the objectives are rewritten, the conclusions should be rewritten based on them. |
Author’s response#6:
Thank you for your comment. In accordance with your comments, we have rewritten the objectives of this study in the Conclusion section to be more appropriate.
p.13 line 319
Few previous studies have reported consistent findings on the association between SPS and gastrointestinal disease symptoms. Moreover, they had small sample sizes, used primarily university student samples, or did not adequately control for covariates. Therefore, to address these issues, the present study examined the correlation between SPS and gastrointestinal disease symptoms using a large sample size with sufficient statistical power, a sample of adults excluding university students, and a wide range of covariates.
p.1 line 13
To fill this gap, using a large sample size with sufficient statistical power, an adult sample not including university students, and a range of covariates, this study examined the association between gastrointestinal disease symptoms as an indicator of physical health and sensory processing sensitivity.
Thank you very much for taking the time to review our paper.

Reviewer 2 Report
In this manuscript, the authors describe the results of a webbased survey in the general population, in which they tried to elucidate the association between (self-reported) sensory processing sensitivity (as a personality trait, not part of sensory processing disorder) and (self-reported) gastro-intestinal symptoms. As they hypothesized on beforehand, they found a positive association: the higher the sensitivity, the more GI-symptoms.
Although the authors feel that their results are the first to show that sensory processing sensitivity (as other personality traits) are not only relaxed to mental health, but also to physical health: they seem to overlook themselves that mental and physical health are clearly intertwined and in interaction with one another, and that particularly GI-symptoms can be part of several (severe) mental disorders, such as depressive disorder, anxiety disorders, somatic symptom disorder etc.etc. It may therefore well be, that they just found an association between HSP and mental health problems all over again.
Methodologically, I have some comments and questions as well.
1. The authors focused on young (predominantly physically healthy) adults, but excluded to some reason ‘students’. Please explain why.
2. It is not fully clear how and why they came from 1651 initial interesten participants to 863 participants in the study. Inclusion criteria are not mentioned in detail. Furthermore, please comment in the discussion on any (assumed or on the basis of missing data analyses) selection effects and the risk of bias.
3. Likewise, please comment in the discussion on information biases, such as reporter bias.
4. It is not clear whether all sociodemographic characteristics in the analyses are as relevant for the association under study, confounding effects seem to be small. Why smoking habits are negatively associated to GI symptoms in almost all analyses deserves some clarifying remarks.
Author Response
**** Reviewer#2 ****
Thank you very much for your constructive comments to improve the manuscript. We strongly agree with all your comments and have revised the manuscript accordingly. Please see our responses below.
|
Comment#1: The authors focused on young (predominantly physically healthy) adults, but excluded to some reason ‘students’. Please explain why. |
Author’s response#1:
As already mentioned in The current study section of the Introduction, prior studies have used only student samples. To address these issues of bias in the existing findings, we recruited a non-student sample as research participants. We have added this point again to the Methods section.
p.3 line 128
To address the issue that existing findings are based primarily on university student samples [6, 15, 37], …
|
Comment#2: It is not fully clear how and why they came from 1651 initial interesten participants to 863 participants in the study. Inclusion criteria are not mentioned in detail. Furthermore, please comment in the discussion on any (assumed or on the basis of missing data analyses) selection effects and the risk of bias. |
Author’s response#2:
Thank you for your comment. We have addressed this issue by adding a new flowchart regarding the sample collection process.
p.4 line 143
Figure 1. Procedures for recruiting participants in this study.
|
Comment#3: Likewise, please comment in the discussion on information biases, such as reporter bias. |
Author’s response#3:
As per your suggestion, we have added the following text to the Limitations section:
p.13 line 312
Finally, the findings of this study are based on data obtained from a web survey and may have been affected by sampling bias, including selection bias and reporter bias.
|
Comment#4: It is not clear whether all sociodemographic characteristics in the analyses are as relevant for the association under study, confounding effects seem to be small. Why smoking habits are negatively associated to GI symptoms in almost all analyses deserves some clarifying remarks. |
Author’s response#4:
Although there was no clear association between covariates and gastrointestinal disease symptoms in our sample, we included covariates that have been reported to be associated with gastrointestinal disease symptoms in previous studies.
For example, a study using a sample older than our sample (40-64 years) reported that smoking habits were positively associated with dyspepsia (Wildner-Christensen et al., 2006). In this regard, smoking habits may not have been a clear risk factor for gastrointestinal disease symptoms because our sample was younger in age Although smoking habits are generally known as a risk factor for H. pylori infection associated with gastrointestinal disease symptoms (Arkkila et al., 2007), a previous study reported a negative correlation between smoking habits and H. pylori infection in young individuals (Ogihara et al., 2000).
[References]
Arkkila, P. E., Kokkola, A., Seppälä, K., & Sipponen, P. (2007). Size of the peptic ulcer in Helicobacter pylori-positive patients: association with the clinical and histological characteristics. Scandinavian journal of gastroenterology, 42(6), 695-701.
Li, L. F., Chan, R. L. Y., Lu, L., Shen, J., Zhang, L., Wu, W. K. K., … & Cho, C. H. (2014). Cigarette smoking and gastrointestinal diseases: the causal relationship and underlying molecular mechanisms (review). International journal of molecular medicine, 34(2), 372-380.
Ogihara, A., Kikuchi, S., Hasegawa, A., Kurosawa, M., Miki, K., Kaneko, E., & Mizukoshi, H. (2000). Relationship between Helicobacter pylori infection and smoking and drinking habits. Journal of gastroenterology and hepatology, 15(3), 271-276.
Wildner-Christensen, M., Hansen, J. M., & De Muckadell, O. B. S. (2006). Risk factors for dyspepsia in a general population: non-steroidal anti-inflammatory drugs, cigarette smoking and unemployment are more important than Helicobacter pylori infection. Scandinavian journal of gastroenterology, 41(2), 149-154.
In the covariates section, we have added the rationale for including these covariates in this study. We have also added references to the cited literature.
p.6 line 183
For example, gastrointestinal disease symptoms are associated with factors such as income [17], alcohol consumption [7], smoking habits [40], BMI [9], allergies [31], previous experience with therapy [23], and physical activity [22].
Thank you very much for taking the time to review our paper.

Round 2
Reviewer 2 Report
Please comment on my major comment:
Although the authors feel that their results are the first to show that sensory processing sensitivity (as other personality traits) are not only relaxed to mental health, but also to physical health: they seem to overlook themselves that mental and physical health are clearly intertwined and in interaction with one another, and that particularly GI-symptoms can be part of several (severe) mental disorders, such as depressive disorder, anxiety disorders, somatic symptom disorder etc.etc. It may therefore well be, that they just found an association between HSP and mental health problems all over again.
Author Response
**** Reviewer#2 ****
|
Comment#1: Although the authors feel that their results are the first to show that sensory processing sensitivity (as other personality traits) are not only relaxed to mental health, but also to physical health: they seem to overlook themselves that mental and physical health are clearly intertwined and in interaction with one another, and that particularly GI-symptoms can be part of several (severe) mental disorders, such as depressive disorder, anxiety disorders, somatic symptom disorder etc. It may therefore well be, that they just found an association between HSP and mental health problems all over again. |
Author’s response#1:
Thank you again for your constructive comments to improve the manuscript. Given that, as you point out, psychological and physical problems are interrelated, the association between SPS and gastrointestinal disease symptoms may partially reflect the association between SPS and psychological health. We have added to the Limitaiton section on this point.
line 314
Finally, it should be noted that although we have shown an association between SPS and physical well-being, given that gastrointestinal disease symptoms are also linked to mental illness, including depressive symptoms, this study may only rediscover the association between SPS and psychological well-being.
Thank you very much for taking the time to review our paper.
